# Analysis of the Consumption of Non-Oncological Medicines: A Methodological Study on Patients of the Ada Cohort

**DOI:** 10.3390/healthcare9091121

**Published:** 2021-08-30

**Authors:** Giulio Barigelletti, Giovanna Tagliabue, Sabrina Fabiano, Annalisa Trama, Alice Bernasconi, Claudio Tresoldi, Viviana Perotti, Andrea Tittarelli

**Affiliations:** 1Cancer Registry Unit, Fondazione IRCCS Istituto Nazionale dei Tumori, Via Venezian 1, 20133 Milan, Italy; giovanna.tagliabue@istitutotumori.mi.it (G.T.); sabrina.fabiano@istitutotumori.mi.it (S.F.); claudio.tresoldi@istitutotumori.mi.it (C.T.); viviana.perotti@istitutotumori.mi.it (V.P.); andrea.tittarelli@istitutotumori.mi.it (A.T.); 2Evaluative Epidemiology Unit, Fondazione IRCCS Istituto Nazionale dei Tumori, Via Venezian 1, 20133 Milan, Italy; annalisa.trama@istitutotumori.mi.it (A.T.); alice.bernasconi@istitutotumori.mi.it (A.B.)

**Keywords:** medicine consumption, defined daily dose, adolescents and young adults, cancer patients, fragility, pain

## Abstract

Cancer patients are identified as fragile patients who are often immunodepressed and subject to secondary diseases. The Ada cohort comprises cancer survivors aged 15–39 years at diagnosis included in 34 Italian cancer registries. This study aimed to analyze the possible excess of non-cancer medicines use on the basis of the medicine database of the Ada cohort. Records of medicines present in the pharmaceutical flows collected by eight Lombardy cancer registries and used by patients with any type of cancer were extracted for the year 2012. Medicine consumption data were processed to assign a defined daily dose value and to evaluate the consumption of medicines belonging to different groups of the ATC (Anatomical Therapeutic Chemical) classification. The values were compared with values in the Lombardy population. Medicine consumption related to 8150 patients was analyzed, for a total of 632,675 records. ATC groups A and C for females and group N for both sexes showed significant increases. Group J for males and group M for females showed intermediate increases, and group H for both sexes showed smaller increases. This method allowed the identification of excess medicine use to reduce cancer therapy side effects and primary disease sequelae in this group of patients.

## 1. Introduction

The Ada project (Adolescent and Young Adult Cancer Survivors in Italy) [1], includes a database of patients who received a cancer diagnosis when aged between 15 and 39 years. In addition to information on the patients and their cancer types, the database lists the sources of the information, which are commonly collected by cancer registries (CRs). Among the most important sources are hospital discharge records, pathology laboratory data, outpatient data, and pharmaceutical prescription records.

The Ada database includes a total of 112,392 records of incident cancer cases between 1976 and 2015 related to 108,777 patients. The records were collected and sent to the database by 31 Italian population CRs and 3 Italian specialist CRs.

In recent years, there has been a significant decrease in the number of hospital discharge records (Figure 1), formerly the main source of epidemiological information. It is therefore useful to evaluate the potential of other sources of information, such as the source considered in the present study: medicine prescriptions.

The pharmaceutical prescription records in the Ada database represent prescriptions issued according to the standards of the Italian National Health System (SSN). They may be subject to full or partial reimbursement according to the following prescription classes: A (life-saving medicines), C (non-essential medicines), and H (medicines for hospital use). Over-the-counter medicines and those prescribed without reimbursement by the SSN are not considered here since their prescription does not follow well-defined protocols, and they are not always reported.

The pharmaceutical prescription records are mainly composed of two data flows:(1)The T flow is dispensed by community pharmacies, i.e., local pharmacies open to the public. This flow includes all medicines [3] distributed by the SSN upon payment of a co-payment or free of charge.(2)The F flow is dispensed by hospital pharmacies or by the local services of the Public Health Agency (ATS, USL or AUSL in Italy). This flow comprises various types of medicines, which may vary from region to region and which in the Lombardy region include:
Innovative hospital medicines;Outpatient medicines;Off-label medicines;Hyposensitizing therapies;Medicines issuable by specialist prescription only;Medicines administered to foreigners with an individual Temporarily Present Foreigner (STP) code;Medicines for rare diseases;Medicines delivered at hospital discharge for the first cycle of care;Medicines distributed by penitentiary institutions;Medicines distributed by Local Public Health Agencies;Medicines administered in hospital to patients with hemophilia;Medicines under risk-sharing agreements;Some blood components;New antiviral medicines for HCV treatment;Others.


### Objectives

The hypothesis underlying the study was that there might be an excess consumption of non-cancer medicines in adolescent and young adult cancer patients. A consequent objective was to analyze and possibly justify the reasons for this consumption. As far as we know, there have been no previous studies analyzing the non-cancer medicine consumption in this specific group of fragile patients [4].

## 2. Materials and Methods

### 2.1. Data Selection

Of the 34 CRs contributing data to the Ada database, 14 provided pharmaceutical dispensation data. A total of 2,328,057 records for the years 1980–2012 were available. Since the highest coverage in the database was for the years 2010 to 2012, we decided to evaluate medicine consumption in 2012, the more recent year. The database included 280,812 registrations for 2012, relative to 12 CRs. To obtain correct and complete terms of comparison despite the absence of an internal standard, we chose 8 CRs in the Lombardy region among these 12 CRs, as their data could be compared with data available on the web (Lombardy Open Data [5]). The data of eight Lombardy population CRs were used for comparison, covering a total of 8,370,359 inhabitants, 4,307,101 females, and 4,063,258 males.

Since the goal was to evaluate and compare the consumption of medicines by adolescent and young adult cancer patients and since Lombardy Open Data allows to select data for the age group between 18 and 39 years, all patients of this age group with previous or recurrent cancer, incident according to the IARC (International Agency for Research on Cancer)-ENCR (European Network on Cancer Registries) criteria for all available years, who had taken medicines in 2012, were selected. For this group of patients, we extracted the medicine dispensation records in the pharmaceutical Ada database for flows T and F and prescription classes A (life-saving), C (non-essential), and H (hospital use).

### 2.2. Data Elaboration

All pharmaceutical records selected were subsequently processed in order to:Calculate the punctual prescription units of medicines for 2012;Group the records by pairing the patients with the marketing authorization numbers (AIC codes) of the medicines they had taken;Add the NDP (number of defined daily doses (DDDs) in the package) to each patient + AIC pair;Calculate the total number of DDDs for each patient + AIC pair.

The analysis involved the assignment for each type of dispensing (package with AIC) of an NDP, which is the result of multiplying the product of the DDDs by the total quantity of active ingredient present in the package.

Some problems were encountered at this point: some AICs lacked an NDP because the regulatory body had not assigned it due to technical impossibility, and several pharmaceutical records had wrong AICs or had been assigned an AIC-unrelated internal code.

After assignment of the NDP, the total DDD was calculated for each grouped record (patient + AIC pair). We then calculated the total DDD for each group of the Anatomical Therapeutic Chemical (ATC) classification system [6].

Since Lombardy Open Data provides only values as grouping by first ATC group (first letter of the ATC code) the values of the Ada data were grouped in the same way.

The DDD/1000 inhabitants/day (DDDid) was then calculated using formula (1) [7]:(1)$$\mathrm{DDD}id=\frac{\mathrm{NDP}\times \mathrm{PP}\;\times 1000}{\mathrm{POP}\;\times \;\mathrm{RD}}$$

Or, in our case, using Formula (2):(2)$$\mathrm{DDD}id=\frac{\mathrm{DDD}tot\times 1000}{\mathrm{POP}\;\times \;\mathrm{RD}}$$
where NDP is the number of DDDs per package; PP is the number of packages prescribed; POP is the total population of the area, re-proportioned by age group 18–39 and sex; RD is the number of reference days, i.e., 365 (1 year); DDDtot is the number of total DDDs, obtained from the sum of the DDDs per ATC group of the grouped records.

The choice of the population in the denominator required special attention. In fact, the cohort whose medicine consumption we analyzed in this study was made up of an atypical set of cases, for which the use of the reference standard population for each age group could lead to misleading results. Therefore, we decided to select a population for the denominator that could best represent the whole, calculating it as a re-proportioning of the total population according to the scheme shown in Table 1. Thus, the population in the denominator was derived as a proportion of the total population covered by the CRs, multiplied by the ratio between the number of patients considered in the study and the total incident cases in one year in the coverage areas of the CRs for all ages.

## 3. Results

From the Ada database, 8150 patients with a cancer diagnosis between 1989 and 2012 and medicine consumption in 2012 were extracted. The group included 4900 females and 3250 males; distribution by life status was available at the most recent follow-up (≤31 December 2017), as reported in Table 2. Most patients were diagnosed quite recently (from one to five years, as shown in Table 2), but there was also a sizeable group with a longer period of observation (up to 15 years).

In Figure 2, Figure 3, Figure 4 and Figure 5, the cases considered are presented by cancer type and sex according to the two main classifications in use: ICCC3 (International Classification of Childhood Cancer [8]) and ICD-10 (the WHO *International Classification of Diseases, 10th edition* [9]).

The processing of the pharmaceutical records selected for the analysis is summarized in Table 3.

Table 4 shows the set of values processed with the relative comparisons, where some excesses of consumption can be noted.

ATC group L, specific to cancer medicines, and groups G + V, which contain medicines used in cancer therapy, were excluded as they are outside the scope of this study.

ATC groups A (alimentary tract and metabolism) and C (cardiovascular system) for females, and ATC group N (nervous system) for both sexes showed significant increases.

ATC group J (anti-infectives for systemic use) for males and ATC group M (musculoskeletal system) for females showed intermediate increases, while group H (systemic hormonal preparations) for both sexes showed smaller increases.

It should be noted that, for both sexes, ATC group R, relating to medicinal products for the respiratory system, presented a decrease in consumption.

The consumptions of ATC groups that showed significant increases was then analyzed by ATC subgroups (to the third and fourth digits). The results, shown in Figure 6, Figure 7, Figure 8, Figure 9, Figure 10, Figure 11, Figure 12 and Figure 13, are expressed as the total number of DDDs prescribed for each group or subgroup and divided into three age groups.

The analysis of the relationship between cancer type, total DDDs of ATC group C, and age did not show significant associations, as shown in Table 5, in which an extract of this sub-analysis is presented. Normally, higher consumption is attributed to the older age group, although for lymphoid and ovarian diseases, the higher consumption is attributable to the intermediate age group (25–32 years).

In Table 6, the consumption of H03A (thyroid preparations) and H05A (parathyroid hormones) medicines is analyzed, comparing patients with thyroid cancer (ICD-10 C73) and patients with other cancers (ICD-10 not C73), who showed substantial differences.

In Table 7, a detail of total DDD for subgroup N07B (medicines used in addictive disorders) is shown.

## 4. Discussion

Cancer patients have a need for more medicines of some of the ATC groups compared with a general patient population, as we found in our pilot study [10]. The cohort of cancer patients considered in this study, which was extrapolated from the Ada database, had particular characteristics, especially given the young age of the patients (18–39 years), but also given the specific peculiarity of the cancer types that most affect these patients [11]. This can greatly influence the consumption of medicines not directly involved in cancer treatment.

We performed analyses of some of the ATC groups that showed consumption increases, evaluating the ratios, gender differences, and clinical aspects of medicine prescriptions. The ATC groups L (antineoplastic), G (sex hormones), and V (miscellaneous) were excluded from the analysis because medicines belonging to these groups are normally used in cancer treatment.

ATC group C (cardiovascular system) showed excess consumption among female patients (Figure 6), to be attributed mainly to blood pressure and lipid regulators in the older age group (33–39 years). As shown in Table 5, this excess was not attributable to any particular cancer type, except as expected in the case of hormone replacement therapy in thyroidectomized patients.

Consumption of ATC group A medicines (alimentary tract and metabolism) was also higher in female patients (Figure 7). The increase was mainly related to subgroup A02 (antacids) and subgroup A11 (vitamins), with substantial consumption of vitamin D, as shown in Figure 8.

Both sexes, albeit with small differences between them, showed increased consumption of medicines belonging to ATC group N (nervous system), with the greatest increase observed for subgroup N07B (substances against abuse, listed in Table 6). In this group methadone (ATC N07BC02) was the agent with the greatest consumption, followed by opioids (N02A), antiepileptics (N03A), and antidepressants (N06A).

H03A (thyroid preparations) represented the subgroup with the highest consumption for ATC group H (systemic hormonal preparations); among female patients there was also a conspicuous peak for the H05A subgroup (parathyroid hormones). Comparative analysis of consumption in these two ATC subgroups between patients with thyroid cancer (ICD-10 C73) and patients with other cancers (ICD-10 not C73) revealed substantial differences, as shown in Table 6.

Additionally, ATC group J (anti-infectives for systemic use) showed a significant increase in consumption, due essentially to antibiotic and antiviral medicines.

### 4.1. Outline of Clinical Pharmacology

In ATC group A (alimentary tract and metabolism), the significantly higher prescription rate among women, almost exclusively in the older age group, was mainly attributable to the A11 subgroup (vitamins) and to a lesser extent to the A02 subgroup (antacids and analogues). Further sub-analysis of A11, also by age group, showed that vitamins are often prescribed in combination, and most prescriptions focus on vitamin A, E, and D combinations, probably because of their antioxidant (E) and anti-osteoporosis (D) effects, and on those of the B complex, probably for their effects on nerve fibers; these three aspects may be useful in counteracting some frequent side effects of cancer chemotherapy. The benefits of vitamin supplementation in cancer patients when not undergoing chemotherapy cycles are less clear and still debated [12,13]. Furthermore, although the increased use of vitamin D in the older age group of females appears consistent with the risk of osteoporosis, especially if they use corticosteroids, it is not clear how gender medicine can justify the other differences. It could be hypothesized that these medicines are used as placebo, and the appropriateness of prescribing them needs evaluation [14].

For ATC group C (cardiovascular system), the gender difference was even more pronounced. Males have a basal consumption in the reference population that is about double that of females, but in the observed cohort, their consumption halved (−49%), while for females a 55-fold increase in consumption was observed. As shown in Figure 6, these are mainly beta-blocking agents (C07) but also real antihypertensives (C02, C09) and anti-dyslipidemics (C10). Additionally, for this ATC group, gender medicine can provide some explanation, given that the hormone blockade in some oncology protocols can have opposite effects in the two sexes as regards the cardiovascular system, but the difference between the sexes was so high that this consideration appears insufficient, and again, the question of prescription appropriateness arises.

In ATC group H (systemic hormonal preparations), females tend to have a 2.5 times higher basal consumption than males, but in the patients of the Ada cohort, consumption almost tripled in both sexes. Disaggregating by ATC subgroup, we observed that females, in whom thyroid neoplasms have a double incidence compared with males (Figure 4 and Figure 5), showed a prescription value of thyroid replacement therapy that was quadruple compared with males. It should be considered that in females there is a higher incidence of autoimmune thyroiditis; however, females have a specificity: unlike males, they also take replacement therapy for iatrogenic hypoparathyroidism, probably in consideration of the fact that they are at greater risk of osteoporosis than males, even before menopause [15].

Additionally, in ATC group J (anti-infectives for systemic use), there was an increase in consumption, modest in females but very marked in males. This increase was attributable to antibacterial medicines (bactericides more than bacteriostatics), antifungals, and antivirals, which are widely used in these patients who are often immunosuppressed as a result of the demanding cancer treatments [16].

In ATC group N (nervous system), females showed an overall triple consumption compared with males, indicating their greater propensity for medicine consumption in general [17,18,19]. The most commonly used medicines by both males and females in our cohort were those classified in the subgroup for substance abuse cessation, but which in these patients are used to treat chronic pain (methadone in 99.5% of cases, buprenorphine in 0.4%). Second in use were other subgroups of specific medicines for the treatment of painful symptoms, which appear relevant and diversified in pathogenesis [20]. These include substances with different mechanisms of action, such as N01B, N02A, N02B, and N02C, but also N06A and sometimes N03A. Last in this category were medicines used, together with non-medicinal techniques, to treat the psychological distress of these patients [21], which in addition to being frequent can in some cases reduce adherence to therapy [22] and sometimes lead to overt mental health issues (N05A, N05B, N05C, and again N06A) [23].

### 4.2. Observations and Limitations

(1)For a more effective analysis it would be preferable to use an internal standard, in the event that records of all prescriptions were available, as the references used (OsMed [24], Open Data, etc.) often have non-compliant characteristics or can limit the calculation needs.(2)It is necessary to have a sufficiently large cohort to avoid a small number of individuals with diseases related to specific ATC groups or subgroups from skewing the results. With a sufficient number of data (not available in this study), the method could make it possible to stratify the excess consumption of medicines based on the time from diagnosis. In this way, possible differences between patients with the most recent diagnoses and other subgroups of patients could be detected, which would allow us to ascertain whether to associate the observed results with the side effects of specific therapies or with the distant outcomes of therapeutic interventions.(3)The groupings of greater detail than the first ATC grouping (first character) are distorting because, although they belong to the same ATC subgroup, the medicines have different characteristics and therefore very different weights in DDD. Evaluating differences in DDD / 1000 inhabitants / day among these medicine subgroups makes no sense. Based on our experience, we have indicated the absolute value of DDD prescribed for some subgroups in order to identify the medicines that most influenced the changes in consumption within the primary group. The evaluation of consumption differences in the ATC subgroups can be performed by comparing the individual subgroups between cohorts with different characteristics, or between a cohort and a reference standard (as expressed in point 1).(4)It is important to carry out appropriate quality control of the pharmaceutical sources, particularly regarding the correct attribution of AIC and ATC codes, which can be problematic when internal codes not corresponding to the official nomenclature are used.

## 5. Conclusions

The analysis of medicine consumption using DDD allows interesting observations to be made on consumption in specific patient populations. Fragile populations, such as the one considered for this study, consisting of cancer patients of the Ada database, show increases in consumption of specific ATC groups and significant differences between the sexes.

These findings can be used for better patient care, and they could be preparatory to actions to prevent and reduce the side effects of therapies and the sequelae of the primary disease, often present in this group of patients.

This technique can be implemented, with appropriate adaptations, for similar analyses in different patient groups. Moreover, if used in the context of the population and in comparison with exposure to environmental agents or adverse events, it can be employed as a sentinel event for monitoring situations of discomfort or suffering.

## Figures and Tables

**Figure 1 healthcare-09-01121-f001:**
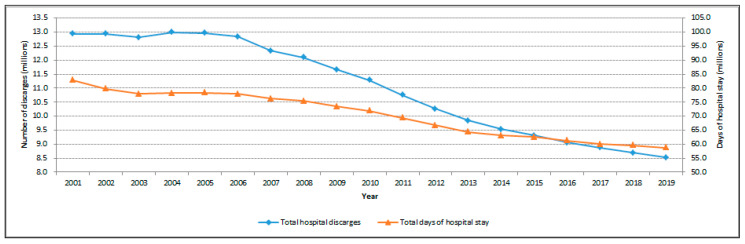
Italian trend of the overall volume of hospital discharges and days of hospital stay. Source: Italian Ministry of Health [2].

**Figure 2 healthcare-09-01121-f002:**
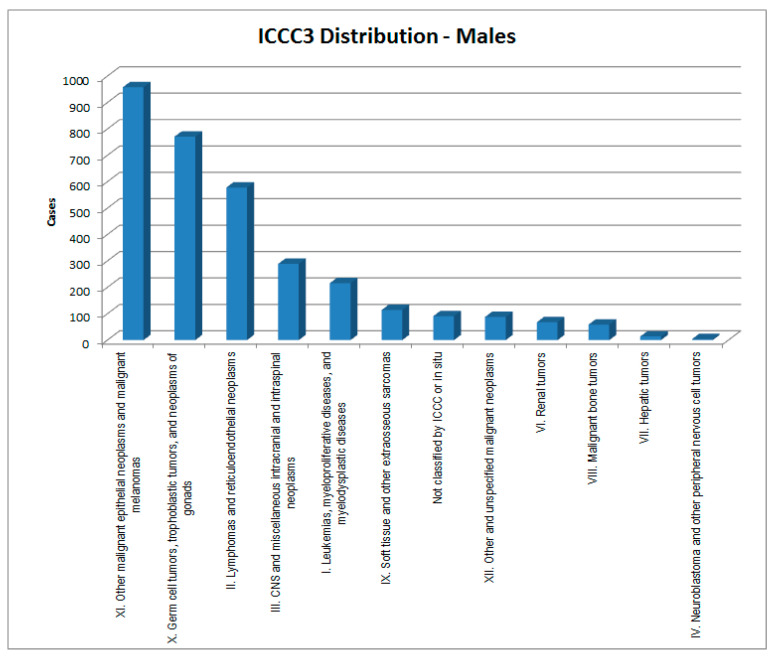
ICCC3 (standard) distribution of cases—males.

**Figure 3 healthcare-09-01121-f003:**
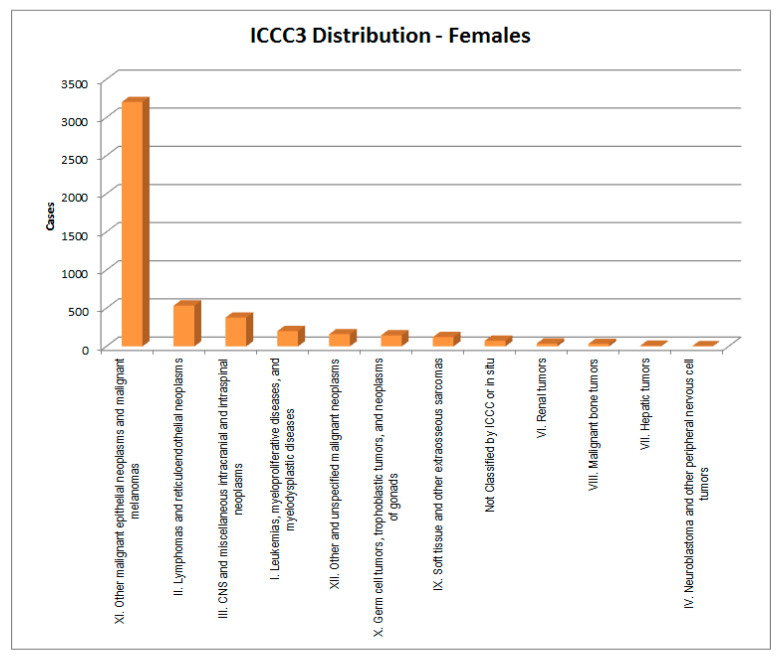
ICCC3 (standard) distribution of cases—females.

**Figure 4 healthcare-09-01121-f004:**
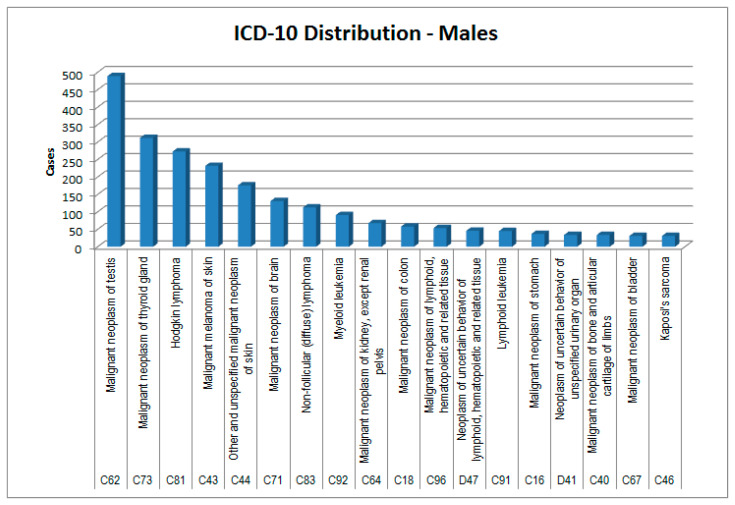
ICD-10 distribution of cases—males.

**Figure 5 healthcare-09-01121-f005:**
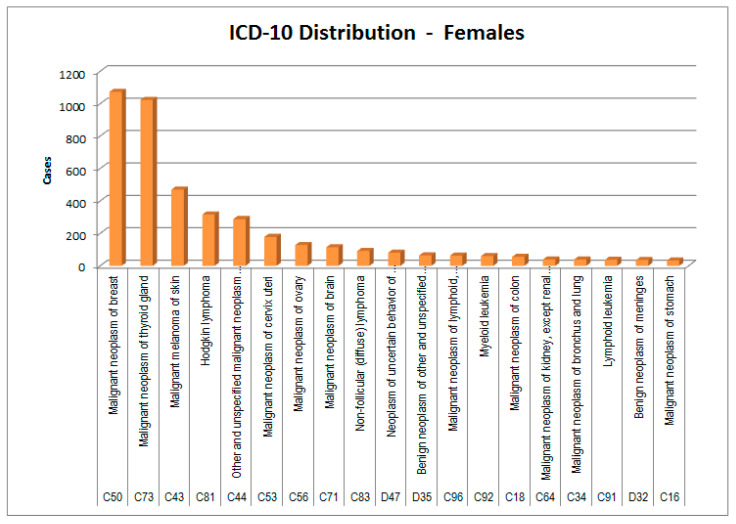
ICD-10 distribution of cases—females.

**Figure 6 healthcare-09-01121-f006:**
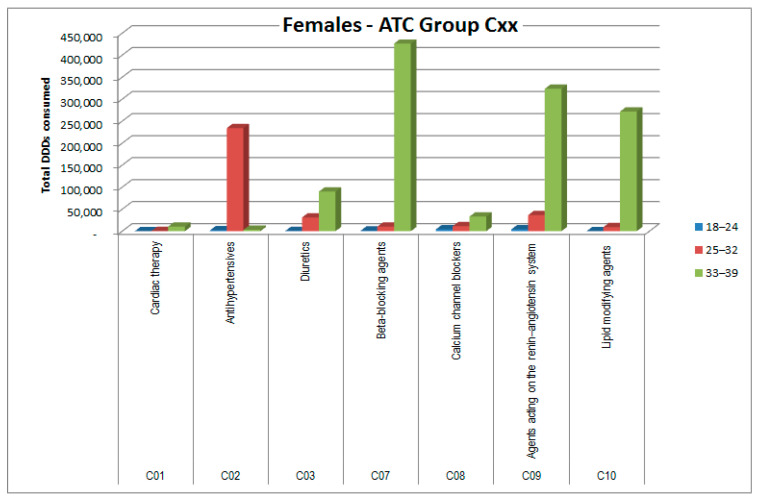
DDD consumption for females, ATC group Cxx.

**Figure 7 healthcare-09-01121-f007:**
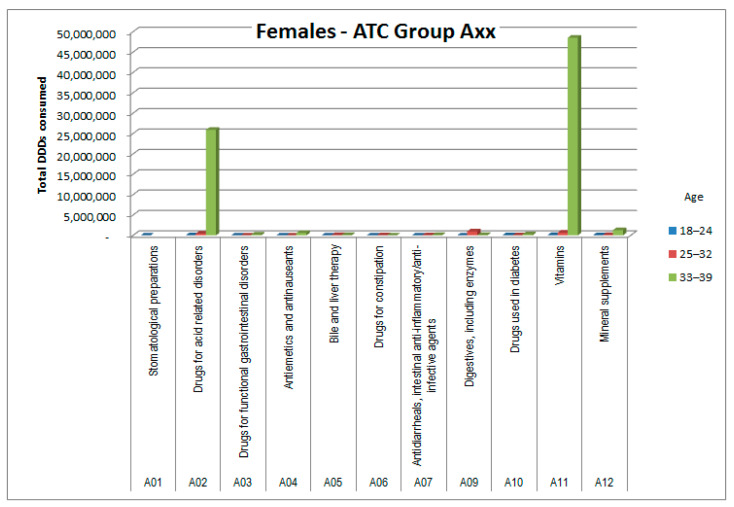
DDD consumption for females, ATC group Axx.

**Figure 8 healthcare-09-01121-f008:**
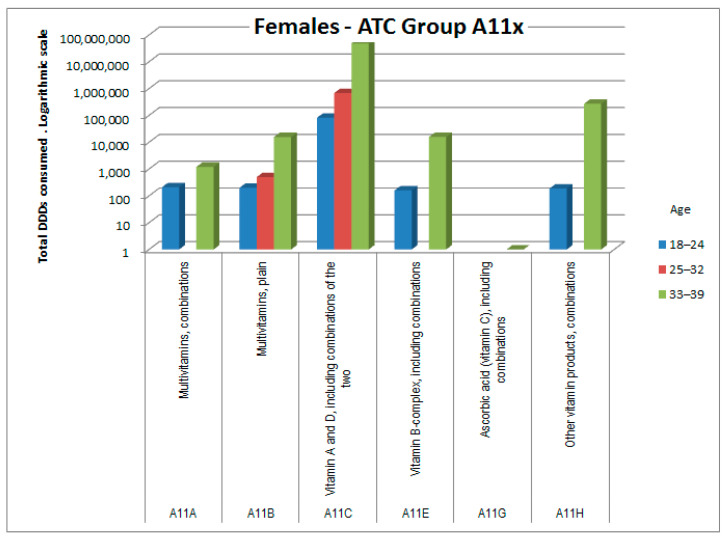
DDD consumption for females, ATC subgroup A11x (vitamins).

**Figure 9 healthcare-09-01121-f009:**
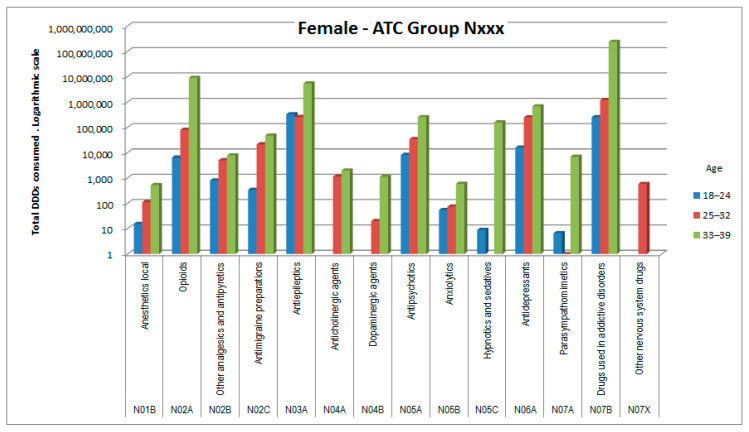
DDD consumption for females, ATC group Nxxx.

**Figure 10 healthcare-09-01121-f010:**
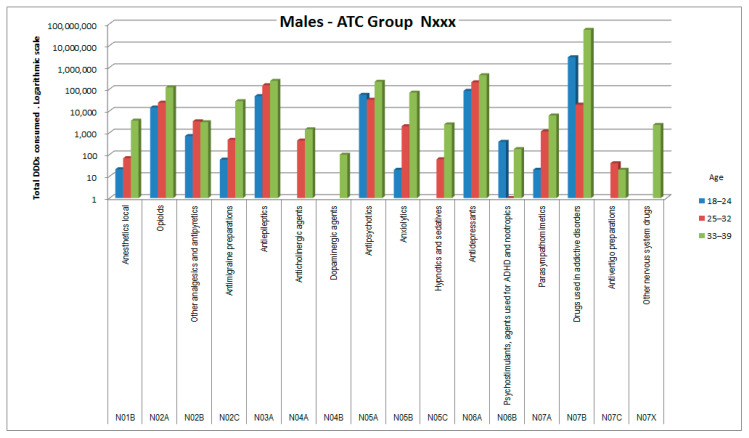
DDD consumption for males, ATC group Nxxx.

**Figure 11 healthcare-09-01121-f011:**
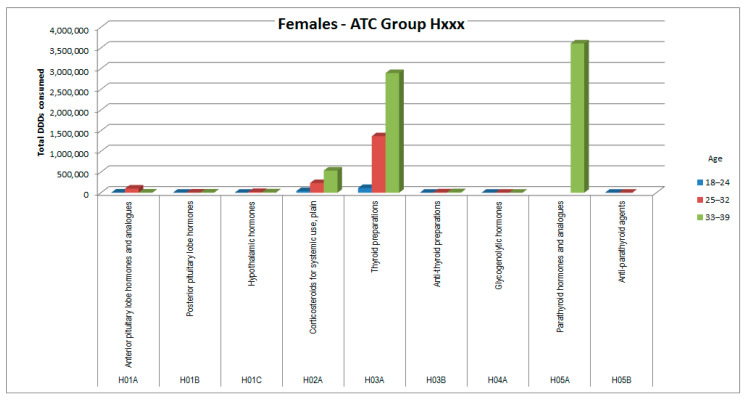
DDD consumption for females, ATC group Hxxx.

**Figure 12 healthcare-09-01121-f012:**
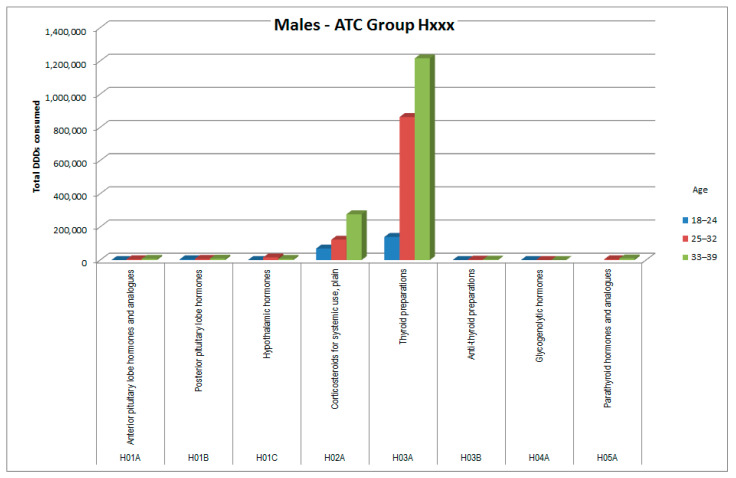
DDD consumption for males, ATC group Hxxx.

**Figure 13 healthcare-09-01121-f013:**
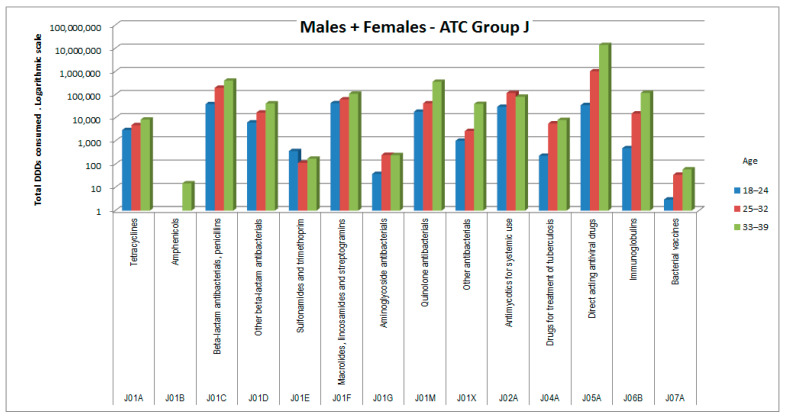
DDD consumption for males + females, ATC group Jxxx.

**Table 1 healthcare-09-01121-t001:** Re-proportioning of the reference population.

	M	F
Total cancer cases in the CR areas for all ages in 2012 (a)	30,671	27,322
Ada cohort patients aged 18–39 years in 2012, with medicine consumption in the same year (b)	3250	4900
Ratio between Ada cohort patients and total CR patients (b/a)	10.6%	17.9%
Total population in the CR coverage area	4,063,258	4,307,101
Reference re-proportioned population for calculation	430,556	772,447

CR = cancer registry.

**Table 2 healthcare-09-01121-t002:** Distribution of cases selected for the analysis, by sex, time elapsed since diagnosis, and life status.

Years Since Diagnosis	0	1–5	6–15	16–25	Total	Alive	Dead	% Dead
Males	387	2092	745	26	3250	3008	242	7.4%
Females	637	3267	965	31	4900	4588	312	6.4%
Total	1024	5359	1710	57	8150	7596	554	6.8%

**Table 3 healthcare-09-01121-t003:** Summary of elaborations on pharmaceutical records extracted from Ada database.

Description	Number of Records	Initial Year, Incident Cases	Final Year, Incident Cases	Notes
Patients with cancer and medicine consumption in 2012, aged between 18 and 39 years	8150	1989	2012	4900 females 3250 males
Records of medicines, 2012	632,675	2012	2012	
Records grouped by patient and AIC (a)	139,931	2012	2012	
Grouped records with invalid AIC	17,931			12.8% of total (a)
Grouped records not connectable to NDP or null NDP	1935			1.4% of total (a)
Total valid grouped records	120,065			

AIC = marketing authorization number; NDP = number of defined daily doses.

**Table 4 healthcare-09-01121-t004:** Comparisons between Ada cohort and Lombardy Open Data for ATC groups in 2012.

			Alimentary Tract and Metabolism	Blood and Blood-Forming Organs	Cardiovascular System	Dermatologicals	Genito-Urinary System and Sex Hormones	Systemic Hormonal Preparations, Excluding Sex Hormones and Insulins	Anti-Infectives for Systemic Use	Antineoplastic and Immunomodulating Agents	Musculoskeletal System	Nervous System	Antiparasitic Products, Insecticides and Repellents	Respiratory System	Sensory Organs	Various
		↓Origin | ATC Group→	A	B	C	D	G	H	J	L	M	N	P	R	S	V
Year 2012 Males, age 18–39	1	Lombardy OPEN DATA 2012 M (DDD/1000 inhabitants/day)	20.18	4.99	23.92	2.63	0.86	6.28	12.6	1.00	3.5	24.86	0.15	22	1.07	0.14
	2	Consumption 2012, Ada DB M patients (DDD/1000 inhabitants/day)	20.04	8.33	12.19	0,32	0.59	17.58	88.40	3564.54	2.13	366.26	0.03	3.33	0.36	2.13
	3	Increase % (2 vs. 1)	−1%	67%	−49%	−88%	−31%	180%	601%	356,354%	−39%	1373%	−80%	−85%	−66%	1421%
Year 2012 Females, age 18–39	4	Lombardy OPEN DATA 2012 F (DDD/1000 inhabitants/day)	19.34	14.6	11.59	1.57	36.54	15.40	16.90	1.76	3.61	27.78	0.62	22	0.75	0.10
	5	Consumption 2012, Ada DB F patients (DDD/1000 inhabitants/day)	327.01	34.80	649.13	0.43	1275.91	51.64	23.08	2444.78	35.82	938.37	0.43	7.90	1.42	1.46
	6	Increase % (5 vs. 4)	1591%	138%	5501%	−73%	3392%	235%	37%	138,808%	892%	3278%	−31%	−64%	89%	1360%
		Excluded for use in cancer therapy														
		100–300%														
		300–1000%														
		>1000%														

DDD = defined daily dose; ATC = Anatomical Therapeutic Chemical classification; M = male; F = female.

**Table 5 healthcare-09-01121-t005:** Total DDDs for ATC group Cxx (females).

ICD-10	Description	Age	Total DDDs
C81	Hodgkin disease	33–39	306,236
C50	Malignant neoplasm of breast	33–39	256,674
C96	Malignant neoplasm of lymphoid, hematopoietic and related tissue	25–32	231,672
C73	Malignant neoplasm of thyroid gland	33–39	164,893
C83	Non-follicular lymphoma	33–39	126,998
C22	Malignant neoplasm of liver and intrahepatic bile ducts	33–39	100,059
C56	Malignant neoplasm of ovary	25–32	27,132
D46	Myelodysplastic syndrome	33–39	25,468
C49	Malignant neoplasm of connective and soft tissue	33–39	24.477
C44	Other and unspecified malignant neoplasm of skin	33–39	21,156
C53	Malignant neoplasm of cervix uteri	33–39	16,263

DDD = defined daily dose; ATC = Anatomical Therapeutic Chemical classification.

**Table 6 healthcare-09-01121-t006:** Analysis of DDDs for thyroid preparations (H03A) and parathyroid hormones (H05A).

ICD10	ATC	TOTAL DDD
C73	H03A	5,617,183
All but not C73	H03A	984,064
C73	H05A	3,606,180
All but not C73	H05A	0

DDD = defined daily dose; ATC = Anatomical Therapeutic Chemical classification.

**Table 7 healthcare-09-01121-t007:** Total DDDs for ATC subgroup N07B (males + females).

ATC Group	Description	Use	Total DDDs
N07BA02	Medicines used in smoke dependence	Medicines used in smoke dependence	30
N07BB	Medicines used in alcohol dependence	Medicines used in alcohol dependence	172,422
N07BC01	Betahistine	Medicines for nausea and vomiting	75,161
N07BC02	Methadone	Severe pain syndromes (dependence)	295,004,425
N07BC51	Buprenorphine, combinations	Buprenorphine, combinations	1,170,013

DDD = defined daily dose; ATC = Anatomical Therapeutic Chemical classification.

## Data Availability

The data will be provided upon request.

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
