# Peer review of "Analysis of the Consumption of Non-Oncological Medicines: A Methodological Study on Patients of the Ada Cohort"

_healthcare, 2021, doi:10.3390/healthcare9091121_

Round 1

Reviewer 1 Report

The authors recruited a large population of cancer patients among the Italian cancer registries to study about the consumption of non-oncological medicines. This might be a very interesting topic for all readers. But I have some concerns about the manuscript preparation and this might be improved after major re-editing, such as

(1). In this manuscript, the authors used a lot of abbreviation without clearly defined at first mention. So, the whole manuscript is very hard to be read and understood.

(2). The data information was few in table 1, 2, and 3. The authors may consider merge the 3 tables into one table.

Reviewer 2 Report

In this manuscript, authors done Analysis of the consumption of non-oncological medicines: a methodological study on patients of the Ada Cohort. In my opinion, some issues should be further address and I hope following comments could be helpful for improving their paper.

  1. Why cancer patients are identified as fragile patients?
  2. Line 105, of the 34 cancer registries... re write this sentence again.
  3. Why the ATC group, the gender difference is more pronounced?
  4. The quality of figures is most important for paper, authors need to put high resolution figures.
  5. I wonder about figure 1, if its not drawn by authors then authors need copyright permission
  6. The manuscript is not well written. There are a number of misused words. I strongly recommend the authors improve it. Also remove grammatical mistakes.

Round 2

Reviewer 1 Report

This revised manuscript has been re-written with much improvement in its quality. And the authors had made appropriate response to the previous reviewers’ comments. I agree that the manuscript is suitable for publication in current form in the journal.